# An Investigation of the Effect of the Work-Function Variation of a Monolithic 3D Inverter Stacked with MOSFETs

**DOI:** 10.3390/mi13091524

**Published:** 2022-09-14

**Authors:** Geun Jae Lee, Yun Seop Yu

**Affiliations:** ICT & Robotics Engineering, Semiconductor Convergence Engineering, AISPC Laboratory, and IITC, Hankyong National University, 327 Jungang-ro, Anseong-si 17579, Gyeonggi-do, Korea

**Keywords:** work-function variation, monolithic 3-dimensional integrated circuits, Monte Carlo simulation, electrical coupling, electrical characteristics

## Abstract

The effect of the work-function variation (WFV) of metal-oxide-semiconductor field-effect transistor (MOSFET) gates on a monolithic 3D inverter (M3DINV) structure is investigated in the current paper. The M3DINV has a structure in which MOSFETs are sequentially stacked. The WFV effect of the top- and bottom-tier gates on the M3DINV is investigated using technology computer-aided design (TCAD) and a Monte-Carlo sampling simulation of TCAD. When the interlayer dielectric thickness (*T_ILD_*) changes from 5 to 100 nm, electrical parameters, such as the threshold voltage, subthreshold swing, on-current, and off-current of the top-tier N-MOSFET and the parameter changes by the change in gate voltage of the bottom-tier P-MOSFET, are investigated. As *T_ILD_* decreases below about 30 nm, the means and standard deviations of the electrical parameters rapidly increase. This means that the coupling and its distribution are relatively large in the regime and thus should be well considered for M3D circuit simulation. In addition, due to the increase in standard deviation, the WFV effect of both the top- and bottom-tier MOSFET gates was observed to be greater than those of only the top-tier MOSFET gates and only the bottom-tier MOSFET gates.

## 1. Introduction

In the semiconductor industry, the research on improving transistor performance and power consumption by increasing integration according to the scaling of the transistor has been conducted [1]. As a future alternative technology for the miniaturization of transistors, a monolithic 3-dimensional (M3D) process technology in which a top-tier transistor is sequentially stacked on a bottom-tier transistor has been proposed [2,3,4,5,6,7,8,9]. The sequential fabrication of multi-transistor layers, M3D offers power, performance, and cost advantages over through-silicon-via (TSV)-based 3D integrated circuits (3DICs) [10,11]. The continuous scaling of transistors can cause process variations, such as line-edge roughness (LER) [12], random dopant fluctuation (RDF) [13], and work-function variation (WFV) [14,15,16,17,18]. LER is the roughness of the edge of the print-line width. As the channel length of the device decreases to less than nanometers, the LER forms channels of different lengths in the channel-length direction of the device, resulting in an electrical characteristic mismatch. RDF is a phenomenon in which the position and density of impurities randomly change during the ion-implantation process of the device. Even with the same number of dopants, the position and density of the dopants in the channel may vary due to the RDF, which may change the electrical characteristics of the device. Due to the continued scale of transistors, high-k/metal-gate (HK/MG) technology has been introduced to overcome the disadvantages of polysilicon. The work-function value of the metal gate electrode used, such as TiN or TaN, has a different value depending on the orientation of the metal particle. Since the size and orientation of the metal particles are randomly determined, the distribution of work-function in the metal gate changes. The change in the WF of the gate metal causes a change in the threshold voltage (*V_th_*), resulting in a mismatch in the electrical characteristics of the transistor [19]. Research has been conducted on characteristics’ variations due to process variations, such as RDF, WFV, and LER for a single device, such as MOSFETS, FinFETs, and gate-all-around (GAA) FETs used in the latest memory and logic devices [20,21,22], but process variations of M3D devices have not been reported. Therefore, it is necessary to study the process variation of the M3D integrated circuit device. As the interlayer dielectric (ILD) between stacked transistors becomes thinner, an electrical coupling in which a gate voltage change of a bottom-tier transistor affects the current of a top-tier transistor is investigated [23], but an electrical coupling considering the process variation of the M3D device is not investigated. Accordingly, it is necessary to analyze the effect of the WFV distribution, which is one of the process variations of the M3D device, on the electrical coupling between the stacked devices.

In this paper, when the gate voltage of the bottom-tier MOSFET is changed from 0 to 1 V in an M3D inverter (M3DINV) in which an N-MOSFET (NMOS) and a P-MOSFET (PMOS) are stacked, the electrical coupling and electrical parameters of the top-tier transistor are investigated for the following three cases: 1. WFV of top-tier MOSFET gate only; 2. WFV of the bottom-tier MOSFET gate only; and 3. WFV of both top- and bottom-tier MOSFET gates. It is investigated through the technology computer-aided design (TCAD) [24] and the Monte-Carlo (MC) sampling simulation of TCAD [25]. Section 2 introduces the method for WFV simulation and Section 3 describes the simulation results for changes in electrical parameters, such as *V_th_*, subthreshold swing (*SS*), off-current (*I_off_*), and on-current (*I_on_*) due to the influence of WFV and electrical coupling of transistors according to ILD thickness (*T_ILD_*). Finally, the conclusion is presented.

## 2. Structure and Method for Work-Function Variation (WFV) Simulation

Figure 1 shows the device structure of M3DINV in which NMOS and PMOS are stacked. The process and structure of this M3DINV is described in detail in [26]. Figure 1a is a cross-sectional view of the M3DINV. To investigate the change in electrical parameters due to the influence of WFV, the channel length (*L*), the gate oxide (*T_ox_*), and the channel width (*W*) were set to 30, 1, and 30 nm, respectively. For NMOS, n-type doped concentration in the source and drain region and the lightly doped drain (LDD) region under the spacer were 10^21^ and 10^18^ cm^−3^, respectively, and the p-type doped concentration in the channel region was 10^15^ cm^−3^. In order to investigate the change in the electrical parameters of the M3D device with respect to WFV, the simulation was performed using the device simulator TCAD. The gate work-function of the stacked devices was set by the MC sampling method using the TCAD design of experiments (DOE) tool [25], and the change in electrical characteristics due to the WFV effect was investigated. Figure 1b shows a schematic diagram of the WFV of the metal grains in the gates of the top- and the bottom-tier MOSFETs. The gate region in the transistor was divided into several segments of the same size, and for each segment, the work functions (WFs) along the grain orientation were randomly determined according to the probability shown in Table 1. The metal gates of NMOS and PMOS were TiN and MoN, respectively, and Table 1 shows the orientations, probabilities, WFs, and average grain sizes (GSs) of the gate materials of NMOS and PMOS [27,28,29]. The TiNs used as the NMOS gate were 4.6 and 4.4 eV according to the <100> and <111> orientations, respectively; the probabilities of the grain orientations were 60 and 40%, respectively; and the average GS was about 5 nm. The MoNs used as the PMOS gate were 5.0 and 4.4 eV according to the <100> and <111> orientations, respectively; the probabilities of the grain orientations were 60 and 40%, respectively; and the average GS was about 15 nm. Assuming a square grain, the number of grains (*N*) in the metal gate area (=*L* × *W*) was defined as (*L*/*GS*) × (*W*/*GS*). With *N* grains, the variation of WF *Φ**_g_* was easily calculated through a binomial distribution model as follows [28]:(1)$$\Phi_{g}=\left(\frac{X_{1}}{N}\right)\Phi_{1}\;+\left(\frac{X_{2}}{N}\right)\Phi_{2}\;+\left(\frac{X_{3}}{N}\right)\Phi_{3}+\cdot \;\cdot \;\cdot +\left(\frac{X_{n}}{N}\right)\Phi_{n},$$
where *X_n_* is a random variable that represents the number of grains with the WF value of *Φ_n_* divided into segment regions in the metal gate, and *Φ_n_* is a randomly designated WF value according to the divided grain regions.

Figure 2 shows the distribution of WF according to the number of grains [28]. The symbols and dotted lines denote reference [28] and the simulation distributions of WF, respectively, and the two distributions show a reasonable agreement within a 10% error. TiN was applied to the NMOS gate metal and MoN was applied to the PMOS gate metal. The mean and standard deviation of the WF distribution of the NMOS metal gate were applied when the number of grains was 36 (= (30/5) × (30/5)), and the mean and standard deviation of the WF distribution of the PMOS metal gate were applied when the number of grains was 4 (= (30/15) × (30/15)). In Figure 2a, the standard deviation increases as the number of grains decreases. This means that the smaller the number of grains, the more affected they are by WFV.

For the MC WFV simulation, according to the flowchart shown in Figure 3, device structure was first created and then current-voltage characteristics were calculated. The left side presented in Figure 3 shows a flowchart for current-voltage characteristics using the TCAD [24]. The device structure including the separation of metal gates with a segmented area was first designed and then the current-voltage characteristics were investigated through the designed device. The right side presented in Figure 3 shows a flowchart for the overall MC WFV simulation of the designed device. The MC WFV simulation started with a TCAD DOE tool [25]. The DOE Internal tool is a process in which the output value changes according to the setting of specified parameters, the input is the mean and standard deviation of the WF, and the output is the WF distribution over all segmented metal regions. The number of simulation samples was first set. Additionally, WF distribution over all segmented metal regions was randomly set using the mean and standard deviation according to the number of grains investigated in Figure 2. Subsequently, the device structure of Figure 1 containing the WF distribution of all the segmented metal regions was generated according to the flowchart on the left in Figure 3. The current-voltage characteristics of the resulting device structure were simulated. The electrical parameters, such as *V_th_*, *SS*, *I_off_*, and *I_on_*, were extracted from the current-voltage characteristics. The simulation was repeated for the number of samples. To investigate the WFV effect of the stacked device, the investigation of changes in electrical parameters was performed in the following three cases: (1) considering the WFV effect of the top-tier MOSFET gate only; (2) considering the WFV effect of the bottom-tier MOSFET gate only; and (3) considering the WFV effect of both the top- and bottom-tier MOSFET gates. The effect of changing *T_ILD_* on the electrical coupling of top-tier MOSFET according to the change in the bottom-tier MOSFET gate voltage was investigated.

## 3. Simulation Results

Figure 4 shows the drain current-gate voltage (*I_DS_-V_GS_*) characteristics of the top-tier NMOS, considering the WFV of both the top- and bottom-tier gates with the method shown in Figure 3, when the gate voltages of the bottom-tier PMOS (*V_BG_*) are 0 and 1 V. The black lines denote the distributions of *I_DS_-V_GS_* characteristics considering the WFV effect with more 400 samples, and the green and red triangles denote *I_DS_-V_GS_* characteristics simulated with the average WF values when *V_BG_* are 0 and 1 V, respectively. The left and right sides of the *y*-axis in Figure 4 show logarithmic and linear scales, respectively, which are indicated by the dotted arrows. Due to the impact of WFV, the distributions of *I_DS_-V_GS_* characteristics are observed, and the threshold voltage shift is observed due to the electrical coupling effect caused by a change of *V_BG_*.

Figure 5 and Figure 6 show the distributions of the electrical parameters of the top-tier NMOS with *T_ILD_* = 10 nm considering the WFV effect of both the top- and bottom-tier gates when *V_BG_* = 0 and 1 V, respectively. Figure 7 shows the distributions of the electrical parameter changes by *V_BG_* between 0 and 1 V of the top-tier NMOS with *T_ILD_* = 10 nm considering the WFV effect of both the top- and bottom-tier gates. The histograms denote the distributions of the electrical parameters and electrical parameter changes obtained by MC sampling simulations of 400 samples. The red lines denote the Gaussian distribution fitted from the MC sampling simulation results, and the extracted means and standard deviations are shown in Figure 8, Figure 9 and Figure 10.

For the three cases where the WFV effect on the top-tier NMOS gate only, the bottom-tier PMOS gate only, and both the top- and bottom-tier NMOS/PMOS gates were considered, the distribution of *I_DS_-V_GS_* characteristics was investigated. The distribution of the electrical parameter changes, such as Δ*V_th_*, ΔSS, Δ*I_off_*, and Δ*I_on_* in *V_BG_* between 0 and 1 V, extracted from the distribution of *I_DS_-V_GS_* characteristics, was investigated according to *T_ILD_* of 5 to 100 nm. Figure 8 shows the means and standard deviations of electrical parameter changes by *V_BG_* between 0 and 1 V according to *T_ILD_* when the WFV effect of the top-tier NMOS gate was only considered. Figure 9 shows the means and standard deviations of electrical parameter changes by *V_BG_* between 0 and 1 V according to *T_ILD_* when the WFV effect of the bottom-tier PMOS gate was only considered. Figure 10 shows the means and standard deviations of electrical parameter changes by *V_BG_* between 0 and 1 V with respect to *T_ILD_* when the WFV effect of both the top- and bottom-tier MOSFET gates was considered. Figure 8a–c and d (Figure 9a–c and d, and Figure 10a–d) show the means and standard deviations of the electrical parameter changes of Δ*V_th_*, Δ*SS*, Δ*I_off_*, and Δ*I_on_*, respectively. The black-filled and red, empty squares denote the means and standard deviations of electrical parameter changes by *V_BG_* between 0 and 1 V, respectively. In Figure 8, Figure 9 and Figure 10, the means and standard deviations of all electrical parameters increase as *T_ILD_* decreases, and especially when *T_ILD_* is over about 30 nm, all the electrical parameter changes are almost constant and relatively very small, and thus the electrical coupling [23,30] and its distribution can be ignored. Although the means of the electrical parameter changes are almost the same in all the three case, their standard deviations for the WFV of both the top- and bottom-tier MOSFET gates considered were greater than those for the other two cases. The variations induced by different process fluctuations, such WFVs of the top- and bottom-tier gates, can be calculated from the following equation [31]:(2)$$\sigma \left(\Delta P_{total}\right)\approx \sqrt{\sigma^{2}\left(\Delta P_{top\_gate}\right)+\sigma^{2}\left(\Delta P_{bottom\_gate}\right)},$$
where *σ*(Δ*P_total_*), *σ*(Δ*P_top_gate_*), and *σ*(Δ*P_bottom_gate_*) are the standard deviations of the electrical parameter changes (Δ*P*s) of *V_BG_* between 0 and 1 V, considering the WFV effect of both the top- and bottom-tier gates as shown in Figure 10, the top-tier NMOS gate only as shown in Figure 8, and the bottom-tier PMOS gate only as shown in Figure 9, respectively. In Figure 10, the blue, empty triangles denote the standard deviations of electrical parameter changes by *V_BG_* between 0 and 1 V, calculated from Equation (2). The calculated standard deviations of electrical parameter changes (blue, empty triangles) show a reasonable agreement with the simulated ones (red, empty triangles) within a 10% error. The means and the measured and calculated standard deviations, shown in Figure 10, are summarized in Table 2.

## 4. Conclusions

The WFV effect of MOSFET gates in an M3DINV structure stacked sequentially with an NMOS and a PMOS was investigated using the TCAD and MC sampling simulation of TCAD. The current-voltage characteristics of the top-tier NMOS in the M3DINV with *T_ILD_* from 5 to 100 nm were simulated when *V_BG_* was 0 and 1 V, and then the electrical parameters, such as *V_th_*, *SS*, *I_off_*, and *I_on_*, and their changes (Δ*V_th_*, Δ*SS*, Δ*I_off_*, and Δ*I_on_*) by *V_BG_* between 0 and 1 V were extracted. Electrical parameters and their changes in the top-tier NMOS were also investigated for the three cases where the WFV effect on the top-tier NMOS gate only, the bottom-tier PMOS gate only, and both the top- and bottom-tier NMOS/PMOS gates were considered. In all three cases, as *T_ILD_* decreased below about 30nm, the means and standard deviations according to the electrical parameter changes rapidly increased. Because the coupling and its distribution were relatively large in the regime, they must be well considered for a circuit simulation and new techniques required. Although the means of the electrical parameter changes at *T_ILD_* ≥ 30 nm were almost the same in all the three case, the results should be considered for the circuit simulation because their standard deviations for the WFV of both the top- and bottom-tier MOSFET gates considered are greater than those for the other two cases. However, in this paper, only the distribution of the electrical parameter changes due to the WFV effect of the MOSFET gates among process variations in M3DINV were investigated. In addition to WFV, it is necessary to investigate the distribution of the electrical parameter changes in M3D devices due to overall process deviations, such as LER and RDF.

## Figures and Tables

**Figure 1 micromachines-13-01524-f001:**
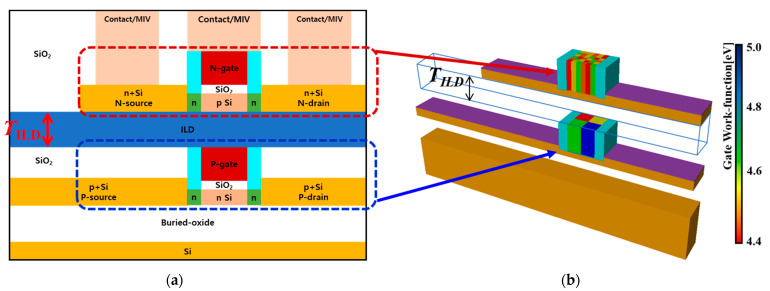
(**a**) Cross-section view of monolithic 3D-inverter (M3D-INV) and (**b**) work-function variation (WFV) of the metal grain on the gate of the top- or bottom-tier metal-oxide-semiconductor field effect transistors (MOSFETs). Here, monolithic inter-tier via (MIV) is shown. The dielectric material of inter-layer dielectric (ILD) is SiO_2_.

**Figure 2 micromachines-13-01524-f002:**
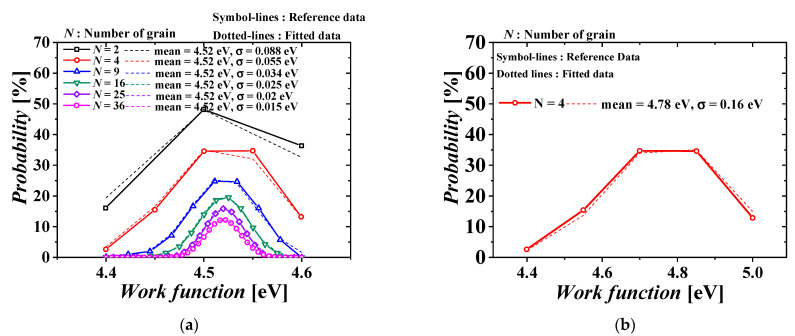
Work-function distributions in metal gates of each tier MOSFET. (**a**) TiN metal gate with the grain size of 5 nm of the top-tier NMOSFET (NMOS) and (**b**) MoN metal gate with the grain size of 15 nm of the bottom-tier PMOSFET (PMOS).

**Figure 3 micromachines-13-01524-f003:**
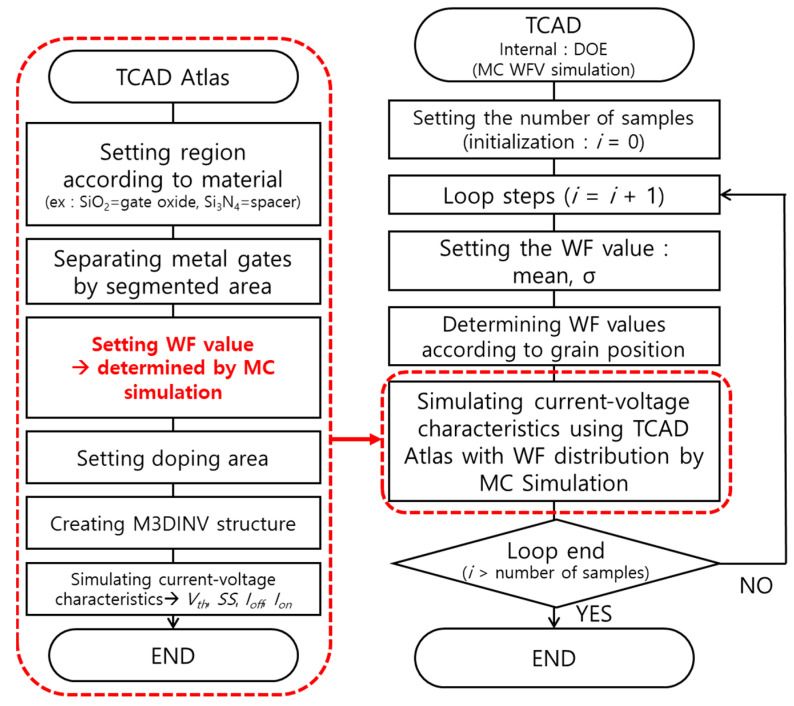
WFV simulation flowchart using technology computer-aided design (TCAD).

**Figure 4 micromachines-13-01524-f004:**
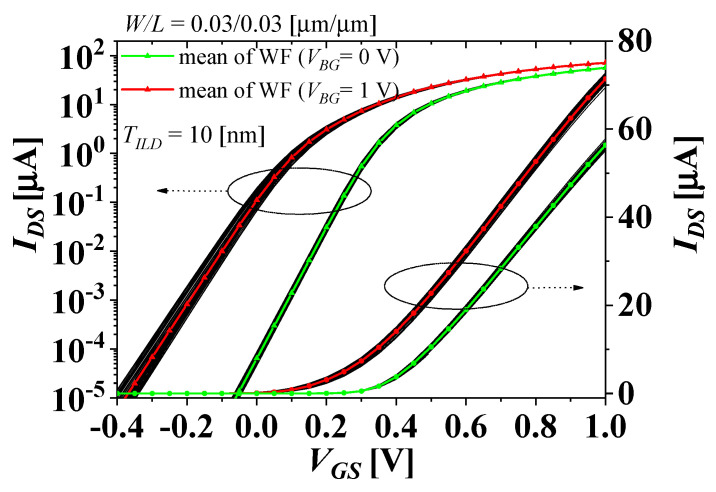
*I_ds_*-*V_gs_* characteristics of the top-tier NMOS considering the WFV of both the top- and bottom-tier gates.

**Figure 5 micromachines-13-01524-f005:**
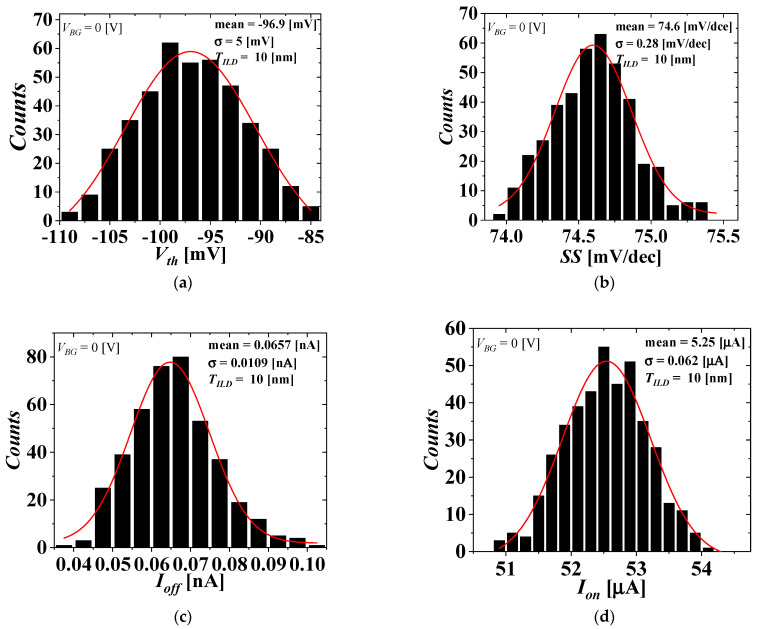
The distributions of the electrical parameters of the top-tier NMOS considering the WFV effects of both the top- and bottom-tier gates when *V_BG_* = 0 V. (**a**) *V_th_*, (**b**) SS, (**c**) *I_off_*, and (**d**) *I_on_*.

**Figure 6 micromachines-13-01524-f006:**
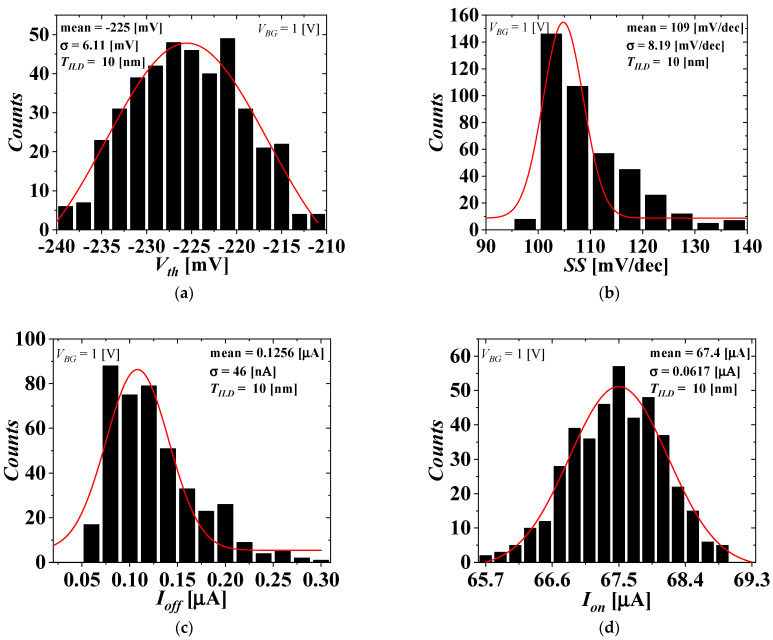
The distributions of the electrical parameters of the top-tier NMOS considering the WFV effects of both the top- and bottom-tier gates when *V_BG_* = 1 V. (**a**) *V_th_*, (**b**) SS, (**c**) *I_off_*, and (**d**) *I_on_*.

**Figure 7 micromachines-13-01524-f007:**
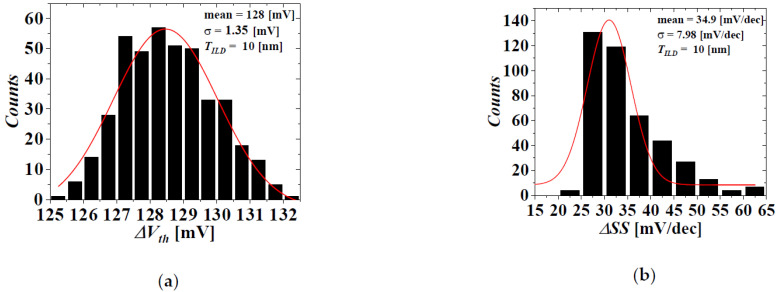
The distributions of the electrical parameter changes by *V_BG_* between 0 and 1 V of the top-tier NMOS considering the WFV effects of both the top- and bottom-tier gates. (**a**) Δ*V_th_*, (**b**) ΔSS, (**c**) Δ*I_off_*, and (**d**) Δ*I_on_*.

**Figure 8 micromachines-13-01524-f008:**
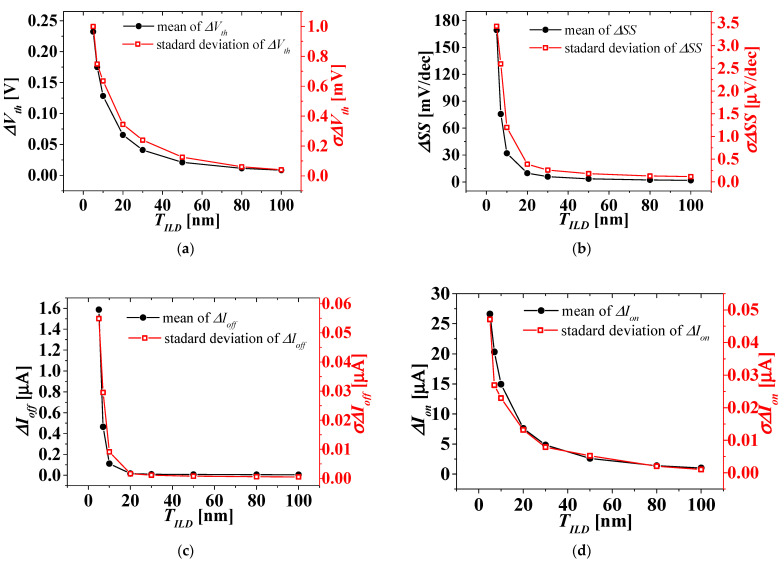
Means (left axis) and standard deviations (right axis) considering WFV on the TiN gate of the toptier NMOS only. (**a**) Δ*V_th_*, (**b**) ΔSS, (**c**) Δ*I_off_*, and (**d**) Δ*I_on_*.

**Figure 9 micromachines-13-01524-f009:**
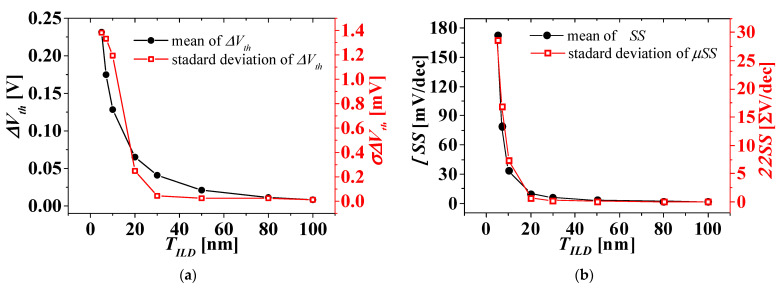
Means (left axis) and standard deviations (right axis) considering WFV on the MoN gate of the bottom-tier PMOS only. (**a**) Δ*V_th_*, (**b**) ΔSS, (**c**) Δ*I_off_*, and (**d**) Δ*I_on_*.

**Figure 10 micromachines-13-01524-f010:**
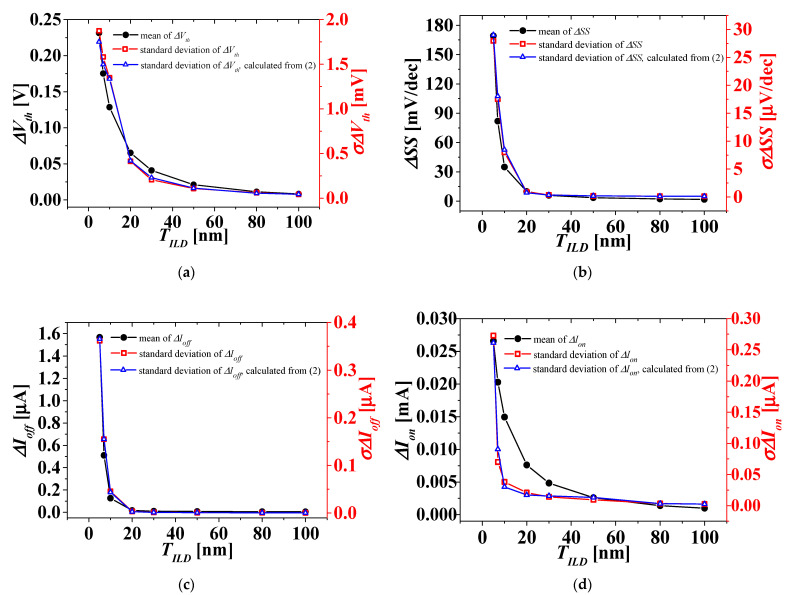
Means (left axis) and standard deviations (right axis) considering both WFVs on the TiN gate of top-tier NMOS and the MoN gate of bottom-tier PMOS. (**a**) Δ*V_th_*, (**b**) ΔSS, (**c**) Δ*I_off_*, and (**d**) Δ*I_on_*.

**Table 1 micromachines-13-01524-t001:** Physical properties of gate materials used for NMOSFET and PMOSFET (orientation, probability, work function, and average grain size).

Device Type	Material	Orientation	Probability [%]	WF [eV]	Average Grain Size [nm]
N-type	TiN	<100>	60	4.6	4.3 [27,29]
		<111>	40	4.4	
P-type	MoN	<110>	60	5.0	17 [27]
		<112>	40	4.4	

**Table 2 micromachines-13-01524-t002:** Means and standard deviations considering both WFVs on the TiN gate of top-tier NMOS and the MoN gate of bottom-tier PMOS. Here, σ* and σ** represent the measured and calculated standard deviations, respectively.

T_ILD_ [nm]	ΔV_th_ [V]			ΔSS [mV/dec]			ΔI_off_ [A]			ΔI_on_ [A]		
	Mean	σ*	σ**	Mean	σ*	σ**	Mean	σ*	σ**	Mean	σ*	σ**
5	0.23144	0.00187	0.00175	169.17	0.02796	0.02902	1.56 × 10^−6^	3.61 × 10^−7^	3.65 × 10^−7^	2.65 × 10^−5^	2.73 × 10^−7^	2.61 × 10^−7^
7	0.17523	0.00158	0.0015	81.84	0.01751	0.01804	5.10 × 10^−7^	1.55 × 10^−7^	1.55 × 10^−7^	2.03 × 10^−5^	6.99 × 10^−8^	9.05 × 10^−8^
10	0.12856	0.00135	0.00134	34.92	0.00798	0.00845	1.25 × 10^−7^	4.54 × 10^−8^	4.37 × 10^−8^	1.49 × 10^−5^	3.81 × 10^−8^	3.01 × 10^−8^
20	0.06525	4.13 × 10^−4^	4.19 × 10^−4^	9.97	9.22 × 10^−4^	7.62 × 10^−4^	1.64 × 10^−8^	3.43 × 10^−9^	2.91 × 10^−9^	7.61 × 10^−6^	2.08 × 10^−8^	1.69 × 10^−8^
30	0.04091	2.06 × 10^−4^	2.30 × 10^−4^	5.91	3.71 × 10^−4^	3.42 × 10^−4^	9.73 × 10^−9^	1.47 × 10^−9^	1.34 × 10^−9^	4.84 × 10^−6^	1.41 × 10^−8^	1.56 × 10^−8^
50	0.02105	1.07 × 10^−4^	1.12 × 10^−4^	3.49	1.87 × 10^−4^	2.01 × 10^−4^	7.01 × 10^−9^	8.39 × 10^−10^	8.33 × 10^−10^	2.60 × 10^−6^	9.42 × 10^−9^	1.31 × 10^−8^
80	0.01129	6.08 × 10^−5^	5.53 x10^−5^	2.23	1.43 × 10^−4^	1.21 × 10^−4^	5.46 × 10^−9^	6.49 × 10^−10^	5.43 × 10^−10^	1.38 × 10^−6^	3.43 × 10^−9^	3.30 × 10^−9^
100	0.00828	4.24 × 10^−5^	4.43 × 10^−5^	1.77	1.17 × 10^−4^	1.17 × 10^−4^	4.71 × 10^−9^	5.37 × 10^−10^	5.27 × 10^−10^	1.01 × 10^−6^	2.47 × 10^−9^	2.53 × 10^−9^

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
