# Peer review of "An Investigation of the Effect of the Work-Function Variation of a Monolithic 3D Inverter Stacked with MOSFETs"

_micromachines, 2022, doi:10.3390/mi13091524_

Round 1

Reviewer 1 Report

Please explain, revise or comment:

  -What is "MIV" in Fig.1?

- Please specify what the ILD permittivity is used for calculations?

- It confuses me: these are indeed microvolts and not millivolts in Figs.5b and 6b?

- DIoff dustributions in Figs. 5,6,7(c) more like lognormal than normal (see e.g. [ doi: 10.1109/TED.2019.2907816.]) Please discuss.

Author Response

Please find an attached file.

Yun Seop Yu

Reviewer 2 Report

The article talks about the WFV in the WF metals due to process variation and its effects on the device electrical parameters in a stacked MOSFET as a function of interlayer dielectric thickness. The article needs to address the following comments,

  1. The article talks so much about WFV, so it must give a brief intro on the need for WF, gate metal, and Vt.
  2. Which material was considered for ILD in the modeling, no clear discussion in the text? Is the material capacitance an important parameter for ILD consideration if so please discuss about it.
  3. A very thin 1 nm SiOx as gate oxide is considered. In this regime, the tunneling current will be very high. Why was it not considered in the study? Also, why wasn’t an HK oxide like HfO2 considered instead of SiOx?
  4. No. of grains is 36 for TiN and 4 for MoN. What is the significance of this number? Why are they different? Some text is needed to clarify it.
  5. Why MC simulations for 400 samples, why not more? what's the significance?
  6. In fig. 4, what is the right y-axis and how is it different from the left y-axis? No mention of it anywhere in the text and figure caption.
  7. In Fig 6,7, in a few figures, the distribution doesn't look Gaussian, but the parameters extracted are using a Gaussian fit. How did the authors validate this analysis? it must be shown.
  8. In the figures, the labels (a-d) of the sub-figures are not explained in the figure caption.
  9. In a real stacked MOSFET device, if there’s variation in the WF of metals, it should be present in both the NMOS and PMOS cases. So, the need for considering WF variation only in NMOS and only in PMOS cases should be addressed.
  10. L187: according to TILD? This statement is not clear.
  11. The mean and SD of all the calculated electrical parameters should be summarized in a table for aiding the readers.
  12. Certain sentences in the conclusion are just a repetition of whatever was said in the simulation results section.
  13. No remarks on what a proposed spec. the limit for the WF metals must be based on the distribution analysis.

Author Response

(The authors gave the same response as above.)
